# When Oxidative Stress Meets Epigenetics: Implications in Cancer Development

**DOI:** 10.3390/antiox9060468

**Published:** 2020-06-01

**Authors:** Álvaro García-Guede, Olga Vera, Inmaculada Ibáñez-de-Caceres

**Affiliations:** 1Epigenetics Laboratory, INGEMM, Hospital La PAZ. 28046 Madrid, Spain; alvarogr@ucm.es (Á.G.-G.); inma.ibanezca@salud.madrid.org (I.I.-d.-C.); 2Experimental Therapies and Novel Biomarkers in Cancer, Instituto de Investigación Sanitaria del Hospital La Paz. IdiPAZ, 28046 Madrid, Spain; 3Department of Molecular Oncology, H. Lee Moffitt Cancer Center and Research Institute, Tampa, FL 33612, USA

**Keywords:** epigenetics, cancer, oxidative stress, miR7/MAFG/Nrf2 axe, chemoresistance, cancer therapy

## Abstract

Cancer is one of the leading causes of death worldwide and it can affect any part of the organism. It arises as a consequence of the genetic and epigenetic changes that lead to the uncontrolled growth of the cells. The epigenetic machinery can regulate gene expression without altering the DNA sequence, and it comprises methylation of the DNA, histones modifications, and non-coding RNAs. Alterations of these gene-expression regulatory elements can be produced by an imbalance of the intracellular environment, such as the one derived by oxidative stress, to promote cancer development, progression, and resistance to chemotherapeutic treatments. Here we review the current literature on the effect of oxidative stress in the epigenetic machinery, especially over the largely unknown ncRNAs and its consequences toward cancer development and progression.

## 1. Introduction

### 1.1. Cancer

Cancer is a generic term defining a wide and heterogeneous group of diseases that can affect any part of the organism. It is considered a multiphasic disease primarily characterized by the appearance of abnormal cells in tissues, with uncontrolled growth beyond their limits and that can invade adjacent organs, disseminating to other parts of the body [1]. According to the World Health Organization, cancer is one of the leading causes of death being responsible for 9.6 million of deaths in 2018 (17% of the total death worldwide). As stated by the World Cancer Report in its edition of 2014, the five most frequently diagnosed tumors among men were lung, prostate, colorectal stomach and liver cancers, whereas among women were breast, colorectal, lung, cervix and stomach cancers [1,2].

Cancer development is a consequence of molecular alterations of genetic and/or epigenetic origin. These can be initiated by the accumulation of genetic changes in DNA that affect the DNA sequence, such as mutations and chromosomal rearrangements, or by modifications in DNA, histones and non-coding RNAs that do not change the original sequence (epigenetic modifications). All of these changes promote the clonal selection of those cells that show a more aggressive phenotype. In 2000, different diseases were first grouped together and collectively referred to as cancer based on their major molecular alterations: self-sufficiency in growth signals, insensitivity to antigrowth signals, evading apoptosis, limitless replicative potential, sustained angiogenesis and tissue invasion and metastasis [3]. Afterwards, in 2009, another five characteristics related to the metabolic, proteotoxic, mitotic and oxidative stress, and DNA damage were also considered when defining cancer [4]. The last revision to the molecular definition of a cancer cell was done in 2011 by Hanahan and Weinberg, who determined new features and defined the presence of a tumor microenvironment developed by the cells along the multiple steps of tumorigenesis [5]. Together, all these properties encompass the most up-to-date definition of the cancer cell.

### 1.2. Epigenetics

Epigenetics is the discipline that studies the inheritable changes of gene expression that are produced by chemical alterations of DNA, histones, and the involvement of non-coding RNAs, rather than changes in the original sequence of DNA. All these changes lead to the remodeling of chromatin to promote or impede gene expression. The silencing of gene expression at the chromatin level is necessary for the normal life of eukaryotic organisms, and it is particularly important in the regulation of biological processes such as the embryonic development, differentiation, or genomic imprinting [6].

There are three major mechanisms of epigenetic regulation described.

#### 1.2.1. DNA Methylation

DNA methylation is the most studied and best known epigenetic mechanism. Generally, DNA methylation is a synonym of genic silencing, as it shapes an inaccessible state of the chromatin for the transcription process. This chemical modification consists of the addition of a methyl group (CH_3_) in carbon 5 of a cytosine located, generally, in 5’-Cytosine-phosphate-Guanine-3’ (CpG) dinucleotides regions. CpG dinucleotide distribution is asymmetric along the genome and its accumulation, preferentially in promoter regions, is called CpG Island (CGI) [7,8]. In the human genome, there are approximately 30,000 unmethylated CGIs that warrant the potentially active configuration of constitutive genes. The methylation pattern of DNA is responsible for cell differentiation; thus, its dysregulation leads to a number of diseases, including cancer [9].

The process of DNA methylation is catalyzed by the DNA-Methyltransferases (DNMTs). There are four types of DNMTs, two involved in de novo methylation of DNA during development (DNMT3A and 3B), one responsible for maintaining the methylation patterns after DNA replication (DNMT1), and one without catalytic site that acts in conjunction with the de novo DNMTs to recruit chromatin remodeling complexes [8,10]. This process, which is essential in embryonic development, is related to the dosage compensation in mammals (X chromosome inactivation) [11,12,13] and genomic imprinting (selective silencing of either maternal or paternal genes) [14]. Although this silencing is mediated by DNA methylation, it is necessary for the involvement of the whole epigenetic machinery for the correct development of the process.

#### 1.2.2. Histone Modifications

Another well-known epigenetic mechanism consists of the chemical modification of histones that are part of nucleosomes in chromatin. Similar to other proteins, histones can undergo posttranslational modifications such as acetylation, methylation, phosphorylation, ribosylation, ubiquitination, sumoylation, or glycosylation (reviewed in [15,16]), whose main function is the regulation of DNA accessibility. In general, phosphorylation [17,18,19] and ribosylation [20,21] promote euchromatin and gene transcription, whereas sumoylation triggers gene silencing [22,23] and ubiquitination can play a dual role [24,25,26]. In the present review, we will mainly address the acetylation and methylation of histones, which consist of the addition of NH_3_ groups through the action of Histone Acetyl Transferases (HAT) and CH_3_ groups by the Histone Methyltransferases (HMTs) in specific residues. Cooperation between DNA methylation and histones modifications induce the recruitment of other chromatin modifiers to promote an active or inactive conformation of chromatin [16] (Table 1).

#### 1.2.3. Non-Coding RNAs

Non-coding RNAs (ncRNAs) are a newly identified group of epigenetic regulators that can fine-tune gene expression without altering the DNA sequence. Prior to the discovery of their regulatory capacity, ncRNAs were considered junk sequences accumulated during evolution, since they occupy regions of the genome with no apparent coding function [27]. However, the Encyclopedia of DNA Elements (ENCODE) project contributed to the classification of non-coding RNAs with previously known functions (ribosomal or transcriptional RNA) [28] but also to the identification of novel ncRNAs [29,30]. Nowadays, transcriptional ncRNAs are classified into small ncRNAs and long ncRNAs (lncRNAs). Small ncRNAs can be divided into microRNAs (miRNAs), PIWI (P-element Induced Wimpy)-interfering RNAs (piRNAs), and small interfering RNAs (siRNAs).

## 2. Implication of Epigenetics in the Development of Cancer

A number of studies have shown that cancer cells experience global changes in chromatin that first affect the whole epigenome through a process of general hypomethylation leading to the genomic instability, and second affect the loss of function by the hypermethylation of specific tumor suppressor genes that regulate the signaling pathways involved in the cell differentiation process, such as *APC*, *GATA*-4, or *p16*, allowing the clonal growth and the abnormal survival of cells [31]. In hematological malignancies (leukemia, lymphomas, and myelomas) mutations in epigenetic modifiers such as *JAK2*, *DNMT3A*, *IDH2,* or *EZH2* are common hallmarks of these diseases [32,33,34]. This leads to alterations of the chromatin structure and the silencing of a number of genes such as *p15*, *RASSF1A* [33,34], *TET2*, *CYP1B1* [35], *PDZD2*, *CSDA* [36], *DAPK1*, *ZAP70* [37], *TERT,* or *TWIST2* [38] to promote the growth of cancer cells in these malignancies. The involvement of DNA methylation and histones modifications in cancer is a well-known event that has been long studied in the last decades [39,40,41,42,43,44]. However, the implication of non-coding RNAs and their regulation in cancer is still largely unknown, although in the past years this knowledge has increased considerably [45,46]. Therefore, in the present section, we will analyze the implication of non-coding RNAs in the development of solid tumors.

### 2.1. microRNAs

microRNAs are a group of non-coding small RNAs (measuring 19–23 nucleotides in length) that regulate gene expression through posttranscriptional regulation without altering their DNA sequence. Regulation occurs through miRNA binding to the 3′-untranslated region (3′UTR) of the messenger RNA (mRNA) of a target gene [47].

miRNAs are transcribed as primary microRNAs (pri-miRNAs) by RNA polymerase II from DNA. Afterwards, this molecule suffers two endonucleasic cuts mediated by the (1) Drosha and the Di George Syndrome critical region 8 gene (DGCR8) protein inside the nucleus [48], and (2) Dicer in the cytoplasm. The Drosha cut generates a hairpin of 60–100 nucleotides that is termed the miRNA precursor (pre-miRNA), which is exported to the cytoplasm by the nuclear membrane transporter ‘Exportin-5′. Once in the cytoplasm, Dicer cuts this structure to generate a double-strand RNA molecule of 19–23 nucleotides. The RISC complex (RNA-Induced Silencing Complex) selects one of the strands, which provokes the degradation of the other one and searches for the homology region in the 3′UTR of a mRNA to block its transcription or favor its degradation [47].

miRNAs were first related to cancer in 2002; those that were downregulated were defined as tumor suppressor miRNAs, such as the miR-15a/16-1 cluster in chronic lymphocytic leukemia [49]. In addition, there is another type of microRNAs, such as the miR-17-92 cluster, whose induction increases cell proliferation, survival, and tumor angiogenesis. The gaining or loss of these miRNAs can increase or decrease the activity of several signaling pathways in cancer cells [50]. Moreover, miRNAs can be involved in epigenetic regulation through the activation or inactivation of DNA-methyltransferases. An example of this is the miR-29 family, which is a well-studied tumor-suppressive microRNA that targets the DNA-methyltransferases *DNMT1*, *DNMT3A* and *DNMT3B* [51] in a number of tumors such as ovarian [52], lung [53], liver [54], melanoma [55], and also hematological malignancies [56,57]. Some well-known tumor suppressors and oncogenes, such as c-MYC or p53, regulate the expression of miRNAs to promote cancer progression. For instance, the oncogene c-MYC activates miR-17, miR-19a, or miR-9 to promote cell cycle progression, inhibition of apoptosis, or metastasis (reviewed in [58]). The loss of p53 also promotes cell cycle progression, cell survival, or stemness phenotypes through the p53-activated miRNAs miR-34, miR-200, miR-15a/16-1 or miR-145 (reviewed in [59]). Although still not much is known about gene and miRNA epigenetic regulation in cancer and their implications for chemotherapy response, another regulatory mechanism of miRNA expression is the DNA methylation of CpGs close to the sequence of the miRNAs of intronic regions or even in regulatory promoter regions of the gene in which they are located [59,60,61].

### 2.2. piRNAs and siRNAs

PIWI-interfering RNAs and small interfering RNAs are groups of small non-coding RNAs of approximately 24–31 and 20–22 nucleotides long, respectively. The main difference between them resides in its processing: siRNAs mature from a double-strand RNA precursors such as miRNAs, whereas piRNAs are processed from single-strand RNA precursors [62]. piRNAs bind to PIWI proteins and siRNAs form the RISC (RNA interfering silent complex) complex. Both piRNA–PIWI [63] and RISC complexes [64,65] can transcriptionally repress gene expression by recruiting the chromatin silencing machinery (methyltransferases and deacetylases) or directly bind to the target mRNA by sequence complementary to induce post-transcriptional gene silencing.

piRNAs were initially discovered a short RNAs in Drosophila that inhibit the expression of the Stellate gene in a germ line [66]. However, follow-up studies showed that piRNA elements are conserved along evolution and also control biological processes in somatic cells (reviewed in [62]). Moreover, recent evidences suggest that piRNAs are related with human cancers. For example, piR-651 [67,68] and piR-55490 [69] promote tumor cell proliferation in lung cancer, piR-36712 [70] and piR-021285 [71] suppress cell proliferation and invasion in breast cancer, piR-1245 induces tumor growth and is a poor prognostic biomarker in colorectal cancer [72], and piR-52207 stimulates tumorigenesis in ovarian cancer [73].

siRNAs were discovered in the 1990s, after the injection of transgenes into Petunia plants [74] and Double-stranded (ds) RNA specific to a genomic region into *C. elegans* [75] that resulted in RNA interference (RNAi) associated gene expression changes. A few years later, a transient silencing with synthetic exogenous siRNA in mammalian cells was observed [76], confirming their existence and their gene-repressive function. Since then, the increasing knowledge about siRNAs has been directed more toward the development of the siRNA as therapeutic tools rather than as biomarkers, which have supposed a great advance for the understanding and therapeutics of cancer disease (reviewed in [77,78]).

### 2.3. Long Non-Coding RNAs

Long non-coding RNAs (lncRNAs) are RNA transcripts of more than 200 nucleotides in length that lack evident open reading frames [79]. lncRNAs are transcribed at lower levels than mRNAs, most of them are poorly conserved along evolution, and their expression seems to be more cell and tissue-specific than mRNAs. As RNA molecules, they show a dual activity that allows them to interact with proteins and other nucleic acids to form complex structures to enhance their regulatory role. Their implication in gene expression regulation is wider than the one exerted by microRNAs; therefore, their regulation is also stricter [80,81].

lncRNAs can be classified according to their chromosomal location and type of regulation regarding their associated coding genes. Those lncRNAs encoded within the sequence of a coding gene are classified as “overlapping lncRNAs”, including sense, antisense, bidirectional, exonic, and intronic lncRNAs. Conversely, lncRNAs located between two genes are termed “long intergenic non-coding RNAs” (lincRNAs) [82,83,84,85]. The action of lncRNAs can act in cis when they regulate the expression of another gene encoded in the 1–300 kb upstream region [84,85]. Trans-acting lncRNAs can also regulate genes that are encoded anywhere in the genome. In addition, lncRNAs whose function is exclusively limited to the nucleus are guiders of chromatin modifying elements—such as DNMTs, Polycomb Repressor Complex (PRC), and/or HATs—to repress or activate the transcription. There is another group of lncRNAs that exert their function in the cytoplasm to regulate the expression of mRNAs and microRNAs in different ways [30,82,86].

The first lncRNAs identified are crucial for the correct embryonic development, since they regulate the chromosome dosage compensation (Xist) and the genomic imprinting and silencing of maternal or paternal genes (*H19*) [11,87,88], which exemplifies the important role of lncRNAs regulating the normal functioning of the cell and the organism. Thus, alterations in lncRNAs contribute to the development and progression of human diseases, including cancer. In cancer, the metastasis-associated lung adenocarcinoma transcript 1 (*MALAT1*), which regulates mRNA splicing, is the most studied cancer-associated lncRNA [89]. The expression of *MALAT1* is upregulated in Non Small Cell Lung Cancer (NSCLC) and ovarian cancer metastatic tumors [90,91], promotes aggressive phenotypes [91,92,93], and can be used as a prognostic biomarker in stage I NSCLC [90]. In addition, MALAT1 is reported to be overexpressed in uveal melanoma [94], melanoma [95], hematological malignancies [96,97,98], and other tumor types [99]. Moreover, recent research have identified a number of lncRNAs such as PVT1 (Plasmacytoma Variant Translocation 1), HOTAIR (HOX Transcript Antisense RNA), GAS5 (Growth Arrest Specific-5), SAMMSON (Survival Associated Mitochondrial Melanoma Specific Oncogenic Non-Coding RNA), CASC15 (Cancer Susceptibility 15), or MEG3 (Maternally Expressed Gene 3) that promote cancer development and progression and can be used as biomarkers of the disease (reviewed in [100,101,102,103]). Given the role of epigenetic mechanisms in cancer development and progression, studying how changes in the epigenetic machinery promote aggressive phenotypes would contribute to the development of new therapeutic approaches and better biomarkers in the clinic.

## 3. Oxidative Stress and Its Effect in the Epigenetic Machinery to Promote Cancer

Oxidative stress is defined as an imbalance between reactive oxygen species production (ROS) and the response of antioxidant enzymes. The major signaling pathway that regulates oxidative stress is NRF2 (Nuclear factor erythroid 2-related factor 2)–KEAP1 (Kelch-like ECH-associated protein 1) [104,105,106,107]. An intracellular ROS increase triggers the activation of NRF2 by KEAP1 inhibition. Nrf2 is translocated to nucleus and dimerizes with other proteins, such as small MAFs (musculoaponeurotic fibrosarcomas), to bind to the Antioxidant Response Element (ARE) sequences in the DNA. The Nrf2–sMAF (small MAF family) dimer works similar to a DNA transcription factor, recognizing AREs and activating several antioxidants genes [108,109,110]. Alteration of the NRF2–KEAP1 pathway is one of the most common and most studied events in cancer regarding pro-oncogenic disorders of oxidative stress. Furthermore, the increase of ROS triggers a multitude of response in different signaling pathways such as MAPK (Mitogen-activated protein kinase) [111], NF-κB (nuclear factor kappa-light-chain-enhancer of activated B cells) [112], STAT3 (Signal transducer and activator of transcription 3) [113,114], or PPARγ (Peroxisome proliferator-activated receptor gamma) [115] that promote antioxidant gene expression, proliferation, and survival in response to oxidative stress, which allow cancer cells to progress. While the activation of these pathways can control antioxidant gene expression independently, all these signaling pathways are also interconnected, highlighting the complex regulatory network that controls the response to oxidative stress.

Increased oxidative stress derived from the augmented metabolism of cancer cells is a common event in many tumor types to promote and maintain its tumorigenic potential. In fact, oxidative stress produces genomic instability and genomic damage that can lead to tumorigenesis [5,116,117]. It has been shown that the hypoxic state of tumor cells increases the oxidative stress situation, which leads to structural and epigenetic changes mediated by the Hypoxia-Inducible Factor (HIF)-1 [118,119]. In another study, continuous exposure to the oxidative stress of non-tumoral kidney cells results in malignant transformation [120]. Oxidative stress leads to an expression imbalance both at the level of histones (HDAC1, HMT1, and HAT1) and of epigenetic regulators (DNMT1, DNMT3a, and MBD4) in these cells. However, the acquired tumorigenic potential of these non-tumoral kidney cells decreased after treatment with the DNA demethylating agent 5-aza-2′-deoxycytidine [120], which supports the notion of an implication of the epigenetic machinery in tumor development. Moreover, several studies indicate a link between glutathione (GSH) metabolism and the control of epigenetics mechanisms at different levels. GSH is an important antioxidant enzyme that intervenes in several biological processes. Alterations in GSH synthesis or GSH depletion produce global DNA hypomethylation that could be due a decrease of S-adenosylmethionine (SAM) [121,122], which is a methyl group donor required for the action of DNMTs and HMTs [122,123]. Before DNMTs and HMTs obtain methyl groups from SAM, it suffers a catalytic transformation from methionine into SAM through the methionine adenosyltransferase (MAT) [122,123]. Both MAT and methionine synthase (MS) are very sensitive to oxidative stress and the balance of GSH synthesis, which explains a low activity of methyl transferase enzymes and a decrease of the genomic methylation level in redox imbalance situations (reviewed in [121,122]. All these changes in the oxidative state of the cell lead to the modulation of the epigenetic machinery at every level and alterations that promote tumor development.

### 3.1. Effect over DNA Methylation

Changes in DNA methylation derived from oxidative stress are mainly due to alterations of the activity and function of DNMTs (see above). Several studies suggest that the induction of oxidative stress mediated by hydrogen peroxide increases the activity of DNMT1 and its binding to the promoters of tumor suppressor genes as *RUNX3* [118]. In addition, hydroxyl radicals promote a global hypomethylation due to the interference in DNMTs–DNA binding capacity [124]. In addition, a high oxidative stress state induces changes in the catalytic cycle of iron and therefore to the inhibition of DNA demethylases of the TET family, thus increasing the levels of DNA methylation [119].

Moreover, high oxidative stress situations induced by ROS production increase 8-hydroxydeoxyguanosine (8-OHdG) levels in some cancer types. 8-OHdG triggers a conformational modification that changes the chromatin active state to chromatin repressive state. Therefore, 8-OHdG could promote tumorigenesis due to changes in the methylation patterns of tumor suppressor genes [125]. In addition, 8-OHdG blocks DNMTs–DNA binding, leading to a global hypomethylation of the genome [126,127].

One element that contributes enormously to the oxidative stress and thus in alterations of the epigenetic machinery is the tobacco smoke [128]. Apart from being a potent carcinogen, tobacco smoke induces high levels of ROS that result in the development of diseases such as the chronic obstructive pulmonary disease (COPD), which also leads to the alteration of the DNA methylation patterns [128,129].

### 3.2. Effect over Histone Modifications

It is also well known that oxidative stress induce changes in the acetylation and methylation of histones as it acts over the enzymes that maintain the chromatin state (see above). Oxidative stress induced by hydrogen peroxide can recruit histone modifiers complexes to promoters of active tumor suppressor genes to inhibit them [118,119]. Despite oxidative stress affecting the posttranslational modifications of histones that regulate the chromatin, it does not act in the same way due to the different sensibility to the oxidative stress of the HMTs, HDMs, and HATs [119].

In a similar manner to DNA methylation, one of the most remarkable changes in histone deacetylases (HDAC), which reduce their activity, is produced as consequence of cigarette smoke [130]. COPD patients show a decrease in HDAC2 activity that increases acetylation in histones H3 and H4 of the *NF-κB* promoter and thus to the dysregulation of proinflammatory genes [128].

In addition, there is evidence of a clear interaction between HIF1α and several HDACs and lysine acetyltransferases (KATs) in hypoxia context. The HIF1-directed transcriptional response appears to be responsible, in part, for the increased stabilization of HIF1α due to the action of HDACs and KATs [131,132].

### 3.3. Effect over Non-Coding RNAs

Similarly, transcriptional regulation mediated by non-coding RNAs is also altered in several ways by oxidative stress, as it has been shown for DNA and histones modifications.

Currently, there are a number of microRNAs whose alteration on their expression pattern is due to changes in the cellular oxidative stress [133,134]. For example, miR-200c is upregulated in epithelial cells as a result of increasing ROS and leads to an increase of apoptotic and senescent cells through the action of its target gene *ZEB1* [135]. Other miRNAs whose expression is induced by transcription factors that are sensitive to increased levels of ROS are miR-27a/b through c-MYC [136], miR-200 and miR-506 through p53 [137,138], and miR-206 through p38 (reviewed in [134]). In addition, the processing of miRNAs from their primary form is regulated by DGRC8–Drosha complexes. Recent reports demonstrate that increased oxidative situations decrease the processing capacity of DGRC8, which relies on Fe(III) for its action, and therefore the downregulation of the corresponding mature miRNA [139,140].

There are several miRNAs involved in the NRF2–ARE detoxification pathway, some due to the direct targeting of NRF2 or its natural inhibitor KEAP1, and others due to the indirect action over genes that regulate this signaling pathway (Table 2). For instance, miR-101 inhibits the expression of NRF2 in breast cancer cells to enhance their sensitivity to oxidative stress and suppress proliferation [141]. miR-432-3p binds directly to the KEAP1 coding region, downregulating it and upregulating the transcription of downstream genes of the NRF2–ARE pathway in esophageal squamous cell carcinoma [142]. miR-7 relieves the oxidative stress of neuroblastoma cells by targeting KEAP1, which promotes an increased expression of NRF2 and the transcription of antioxidant genes such as *HO-1* and *GLCGM* [143]. miR-200a binds to the KEAP1 3′UTR sequence leading to the degradation of its mRNA in breast cancer cell [144], hepatocellular carcinoma cells [145], and esophageal squamous cell carcinoma cells [146] (Figure 1). However, further research is needed to fully understand the complex regulatory network between miRNAs and the KEAP1/NRF2 axis.

On the other hand, and despite that lncRNAs are a relative recent discovery, the alteration of these non-coding RNAs have also been linked to oxidative stress. The lncRNA “nuclear lung cancer associated transcript 1” (*NLUCAT1*) is upregulated in hypoxia in lung cancer cells through HIF2A, NF-κB, and NRF2 transcription factors [162]. This lncRNA promotes oncogenic abilities and cell survival in the presence of cisplatin treatment probably by upregulating the expression of antioxidant genes (*ALDH3A1*, *GPX2*, *GLRX*, and *PDK4*) through a mechanism still not understood [162]. In hepatocellular carcinoma cells, cell death produced by erastin, a strong inductor of ferroptosis, is mediated by *GABPB1-AS1* lncRNA [163]. This lncRNA blocks *GABPB1* mRNA recruitment to polysomes to decrease its protein expression and reduce the transcription of peroxidase proteins, which in turn increases ROS production and cell death [163]. The exposure of renal cell carcinoma cells to H2O2 to induce oxidative stress leads to a downregulation of the lncRNA ‘secretory carrier membrane protein 1′ (*SCAMP1*) and to apoptosis of the cells [164]. In this situation, *SCAMP1* acts as a competitive endogenous RNA (ceRNA) with *ZEB1* and *JUN* to sequester miR-429. Thus, the downregulation of *SCAMP1* increases the availability of miR-429 to target *ZEB1* and *JUN* and this way decreases cell viability [164]. This mechanism of “sponging” microRNAs through lncRNAs is a common feature to regulate gene expression by which RNA molecules sequester microRNAs to prevent them for targeting other RNAs. For instance, the expression of *HULC* and *H19* is triggered by oxidative stress to upregulate Interleukin-6 (*IL-6)* and the C-X-C chemokine receptor type 4 (*CXCR4)* sequestering Let-7a/b and miR-372/-373, respectively, to promote cholangiocarcinoma migration and invasion [165]. In multiple myeloma, *MALAT1* positively regulates NRF1 and NRF2 through the transcriptional inhibition of *Keap1* by EZH2 [96], which is a component of the PRC that induces H3K27me3 [166] through a mechanism still under study. In addition, *MALAT1* and EZH2 also interact to inhibit miR-29 in this disease [97], which is interesting due to the negative regulation of Keap1 through miR-29 [167] and its implication in the regulation of ROS [168], as it has been shown in other diseases. While all these reports clearly describe oxidative stress responses mediated by lncRNAs, as summarized in Table 3, a detailed regulation of these non-coding RNAs by reactive oxygen species and their role in cancer development and progression is still under study.

## 4. Chemotherapy-Induced Oxidative Stress: Epigenetics and Cisplatin Resistance

One of the main problems associated with cancer treatment is the development of resistance to different treatments. This resistant state in cancer cells can be explained by intrinsic and/or acquired mechanisms of desensitization to the drug’s action. There are several ways proposed by which cancer cells develop drug resistance such as cellular plasticity, clonal selection due to pharmacological stress, signaling pathways plasticity, or the epigenetic mechanism, among others.

### 4.1. Cisplatin Resistance Mechanisms

Cisplatin (CDDP) is the first-line chemotherapeutic treatment in a multitude of tumors. Cisplatin is an alkylating agent that intercalates in DNA strands, forming adducts that cause genomic damage [171,172,173,174]. Moreover, one consequence of cisplatin administration is the production of high levels of ROS and nitrogen (RNS), which increase the oxidative stress in tumor cells that can promote alterations in the epigenetic machinery [175,176,177]. Although cisplatin is a potent inductor of cell apoptosis due to the increased intracellular oxidative imbalance, amongst other functions, the main limitation of its use is that the disease almost invariably progresses to a platinum-resistant state.

There are a number of events underlying the phenomenon of cisplatin resistance in cancer. One such event consists of alterations in DNA repair mediated by the activation of the epidermal growth factor receptor (EGFR) pathway resulting in cellular proliferation or the nuclear interaction of the receptor with proteins responsible for rejoining double-strand breaks [170]. Other proposed mechanisms include the reduced intracellular accumulation of cisplatin due to the overexpression of molecular ABC transporters that act by transporting the drug out of the cell, the activation of transcription factors that induce cell proliferation, the increase of antiapoptotic proteins, such as BCL-2, and alterations in the epigenetic regulatory machinery, mostly in DNA methyltransferases that modify the expression of a number of genes [172,178,179,180] (Figure 2). Despite the efforts to unravel the specific mechanisms for developing cisplatin resistance, it seems to be a multifactorial effect that includes several of the above-named mechanisms. Thus, understanding these molecular mechanisms of resistance development is a crucial step to improve the treatment of the disease.

### 4.2. DNA Methylation and Histone Modifications

As we mentioned above, cisplatin-resistance development is related to the activation of DNMTs, whose activity can be modified due to the oxidative stress produced by cisplatin treatment. The process of DNA methylation in tumor cells leads to the silencing of specific genes that are crucial in normal conditions for the correct functioning of the cells. This silencing can be direct, if hypermethylation occurs in the promoter region of the gene that becomes inactive, or indirect in those cases where hypermethylation is on regulatory regions of coding and non-coding genes, silencing them and therefore increasing the expression of their target genes.

In fact, the transcriptional silencing of some genes, such as the arginine–succinate–synthetase, mediated by the hypermethylation of the CpG dinucleotides of its promoter, is a frequent epigenetic event associated to the development of cisplatin resistance in ovarian cancer [181]. Other studies describe the loss of expression of *IGFBP-3* (Insulin-like Growth Factor Binding Protein-3) in NSCLC as an effect of cisplatin administration. The silencing of this gene is produced by the hypermethylation of its promoter region in cisplatin-resistant cancer cells and leads to the development of cisplatin resistance through the activation of the Insulin-like growth factor receptor (IGF-IR)/AKT pathway [182,183]. There is increasing knowledge about the genes whose promoters are hypermethylated in cancer and that are related to cisplatin resistance as a consequence of the epigenetic silencing that they are suffering. However, the number of coding and non-coding genes identified up to date remains limited [184].

Alterations in the expression in histones deacetylases and demethylases can also contribute to the development of a cisplatin-resistant state in some tumor types. An example of this occurs in NSCLC in which the increased expression of these enzymes, the histone deacetylase-6 (HDAC6) specifically, decreases sensitivity to cisplatin through the reduction of apoptosis and prevention of DNA damage [185]. On the other side, oxidative stress induced by cisplatin administration leads to changes in histone demethylases, altering the methylation pattern of histones and being a silencing mechanism in cancer [183]. Moreover, cisplatin and other chemotherapeutic regimens also promote ROS induction and ubiquitination of the histone variant H2AX, which determine the sensitivity to the drug [186], reinforcing the importance on histones modifications in redox balance.

The interest of studying the link between epigenetic modifications of non-coding RNAs regulatory regions and the development of platinum-resistant phenotypes has increased in the recent past years. Therefore, in the next sections, we will discuss how alterations of non-coding RNAs promote resistance to cisplatin treatment.

### 4.3. microRNAs

One of the regulatory mechanisms of miRNAs expression is their silencing by the methylation of their regulatory regions, thereby increasing the expression of their target genes [45,187]. One of the first miRNAs identified to be regulated by this mechanism was miR-200c, whose hypermethylation of its regulatory region is responsible for the downregulation of this miRNA and therefore the subsequent development of resistance to chemotherapy in NSCLC cell lines [188]. The silencing of miR-493 by DNA hypermethylation is also involved in promoting cisplatin resistance in lung cancer cells by increasing the expression of its target gene *TCRP1* [189], which is a gene linked to cisplatin resistance in lung, ovarian, and tongue cancer cells [190,191].

There are also a number of miRNAs that regulate the expression of KEAP1/NRF2 to control cisplatin response. miR-144-3p is downregulated in lung cancer to increase *NRF2* expression and promote better cell survival in the presence of cisplatin treatment [159]. In cisplatin-resistant hepatocellular carcinoma cell lines, miR-340 is downregulated, which increases NRF2 activity, thus promoting resistance to the treatment [160]. In addition, miR-141, which targets *KEAP1*, is upregulated in ovarian cancer cell lines to promote cisplatin resistance [161]. Although the authors of this study conclude that NF-KB, and not the NRF2 pathway, is responsible for the observed phenotype, it might be worth studying the miR-141/KEAP1/NRF2 axis due to the known role of NRF2 in promoting cisplatin resistance in other tumor types.

#### miR-7/MAFG/ROS axis

Interestingly, some miRNAs can target the binding partners of NRF2 to affect cisplatin response. An example of this is the miR-7/*MAFG*/ROS axis that leads to cellular and biological responses that promote tumor development and progression in different tumor types. The main epigenetic mechanism involved in this aberrant alteration is the DNA methylation at the CpG Island surrounding the chromosomal location of miR-7 that has been associated with cancer progression [150]. In this study, the authors worked with cellular models of cisplatin resistance established after the chronic exposure to the cisplatin treatment of initially cisplatin-sensitive NSCLC and ovarian cancer cell lines [150,192]. These cisplatin-resistant cell lines share a common downregulation of miR-7 expression, compared to the parental sensitive cells, due to the hypermethylation of a CpG island located on its promoter [150]. Most importantly, lung adenocarcinoma patients harbor a hypermethylated miR-7 promoter when compared to healthy controls [193]. In addition, these patients show lower overall and progression-free survival rates, suggesting that the hypermethylation of miR-7 could be involved in the early establishment of the disease [193]. In fact, miR-7 hypermethylation constitutes a molecular event that accounts mainly at the expense of the emphysema patients, as analyzed from a huge COPD cohort. Patients with emphysema phenotype are considered the group of COPD patients that have a higher risk of developing lung cancer than other COPD phenotypes [194], indicating that the silencing of miR-7 through DNA hypermethylation might be one of the main determinants explaining the higher incidence of lung cancer in these patients [195]. Similarly, studies in ovarian cancer have also shown that the hypermethylation of miR-7 is associated with a worst response to platinum-derived chemotherapy, which is indicated by lower overall survival and progression-free survival rates when miR-7 promoter is methylated [150]. One of the effects of miR-7 silencing in cisplatin-resistant lung and ovarian cancer cell lines is the upregulation of its target gene MAFG (Musculoaponeurotic Fibrosarcoma Oncogene Family, protein G) [150,196] (Figure 3).

MAFG is a bZIP transcription factor that belongs to the small MAF family (sMAFs) of proteins. The sMAFs harbor a basic region motif for DNA binding and a leucine zipper motif for dimerization and are thought to have compensating roles. sMAFs can homodimerize with themselves or heterodimerize with other bZIP transcription factors such as the Cap’N’Collar (CNC) family or the Bach family to activate or repress gene expression in response to oxidative stress. Therefore, miR-7 silencing promotes low levels of ROS in cisplatin-resistant cancer cells due to the upregulation of *MAFG*. In fact, cisplatin-resistant NSCLC cell lines show lower ROS levels than the sensitive counterpart in response to cisplatin treatment (Figure 3). This effect can be overcome by sequestering MAFG through specific aptamers against the protein, as it results in a decrease of cell viability due to an increase of ROS production. In fact, knockout (KO) experiments in mice have shown that the small Mafs are essential for the activation of ARE-dependent genes. MafG KO mice showed mild thrombocytopenia and motor ataxia that improved with age, while MafK and MafF KO mice did not show an apparent phenotype [197], suggesting that the role of MafG is not well compensated by the other sMafs. In addition, the double KO MafG/MafF compromised the expression of *NQO1* and other oxidative stress-response genes [110]. Moreover, other authors have described the essential role of sMafs in regulating the expression of cytoprotective genes [110]. Importantly, ChIP-seq experiments to identify the binding sites of NRF2 and MAFG have demonstrated that, when heterodimerized, they can regulate genes involved in antioxidant and metabolic networks such as *NQO1*, *HMOX1*, *IDH1*, *PGD,* and *G6PD*, amongst others [109].

Despite sMafs have been associated with cellular response, little is known about their implication in human pathologies. However, there are a number of reports that relate the overexpression of MAFG to cancer development and progression. Several studies demonstrate that MAFG promotes aggressive phenotypes in hepatic tumors [198,199,200], bladder cancer [201], and colon cancer [202,203].

In addition to its role in actively repressing or promoting gene expression, MAFG has recently been involved in the regulation of the methylator phenotype in colon and melanoma [204,205]. In both studies, hyperactivation of the MAPK pathway derived by oncogenic BRAF^V600E^ or EGFR^G719S^ prevents the ubiquitination of MAFG by phosphorylation of the S124 through ERK. This posttranslational modification leads to the formation of a protein complex orchestrated by MAFG stabilization and formed by BACH1, CDH8, and DNMT3B. The activation of these complexes leads to the increased methylation of CpG islands located in the tumor suppressive genes of both melanoma and colon cancer [204,205]. Therefore, the implications of MAFG in regulating gene expression are more complex than initially thought and worth studying in the future in order to identify potential biomarkers and therapeutic targets. Most importantly, since the KO mice of MAFG did not show a strong phenotype [197], targeting MAFG as a therapeutic approach, as it has been previously done to restore cisplatin sensitivity in lung cancer cells through DNA aptamers [196], provides promising results and potential therapeutic strategies in the future.

Apart from being stabilized by a posttranslational regulation, MAFG expression is also regulated at different levels. The ChIP-seq experiment mentioned above also confirmed that *MAFG* transcription can be regulated by NRF2 [109], which is an observation that was previously reported in mouse [206], suggesting a regulatory feedback between MAFG and NRF2 in response to oxidative stress in order to cope with high levels of ROS. However, MAFG can also be regulated at the post-transcriptional level by microRNAs. Although the coding sequence of MAFG is relatively small, the 3′UTR is over 4000 nucleotides long and over than 50 miRNAs are predicted to regulate MAFG. Some reported examples include miR-218 in smoking-induced disease processes in lung [207] and miR-7 in NSCLC and ovarian cancer cells [150].

Interestingly, as we mentioned above, miR-7 can also regulate the expression of genes that promote or reduce oxidative stress in the cell. One example is the regulation of *KEAP1* in neuroblastoma cells [143]. In these cells, miR-7 relieves oxidative stress by targeting *KEAP1*, which promotes an increased expression of NRF2 at the posttranslational level, which will promote the transcription of antioxidant genes such as *HO-1* and *GLCGM*. Besides, miR-7 can regulate the expression of proteins that promote oxidative stress situations. A-synuclein is a protein involved in synaptic activity through the regulation of vesicle docking, fusion, and neurotransmitter release [208,209,210]. One of the hallmarks of Parkinson’s disease is the aggregation of a-syn in neurons to cause an intracellular toxic burden. This promotes mitochondrial dysfunction, the decreased activity of the superoxide dismutase 1 (*SOD1*), and low GSH, which in turn results in increased ROS production [211,212]. miR-7, whose expression is mostly present in neurons, can reduce a-syn expression by targeting the 3′UTR region of the mRNA, protecting the cells against the a-syn-mediated proteasome impairment and susceptibility to oxidative stress [209]. In addition, the role of miR-7 regulating a-syn has been studied in the ischemic brains of rodents. In this case, cerebral ischemia in rodents decreases miR-7 expression, which induces α-Syn expression and promotes cell death. Interestingly, the administration of an miR-7 mimic in pre-ischemic rats has a neuroprotective effect and decreased brain damage in post-ischemic brains, while these effects were lost in a-syn-/- mice [208]. Although these reports are not focused on cancer, due to the important role of miR-7 as a tumor suppressor (reviewed in [213]), it would be worth studying its role in regulating oxidative stress through these proteins in cancer.

### 4.4. Long Non-Coding RNA

Different CpG island methylation patterns have also been described between different types of lncRNAs depending on their chromosomal location in several tumor types as a result of chronic treatment with cisplatin [112]. However, the specific role of these silenced lncRNAs in cisplatin resistance remains unknown. Recent evidence suggests that lncRNAs are involved in chemoresistance to various anticancer therapies. One example is HOTTIP, an lncRNA regulating 5′ HOXA gene transcription, which has been associated with cell proliferation, invasion, and chemoresistance in osteosarcoma, liver, and pancreatic cancers [113,114]. The lncRNAs UCA1 and ROR have been associated with the resistance of cancer cells to platinum-based treatments in bladder [115] and nasopharyngeal cancers [116], respectively. In addition, some miRNAs are regulatory intermediates between UCA1 and oxidative stress produced by cisplatin [117] or arsenic treatment [96] in which UCA1 acts as ceRNA with other microRNA target genes. In this context, UCA1 upregulation promotes a competing situation for miR-495 and miR-184 with NRF2 [117] and ‘oxidative stress induced growth inhibitor 1′ (OSGIN1) [96], respectively, to upregulate their expression and promote a better survival to drug treatments of NSCLC cells [117] and **hepatocellular carcinoma** (HCC) cells [96]. Although these reports show the implication of lncRNAs in promoting drug-resistant phenotypes, the specific mechanism by which lncRNAs regulate chemotherapy resistance is yet a novel field for exploration.

## 5. Conclusions and Future Directions

The relevance of epigenetic mechanisms in cancer has increased in the last years considerably. DNA methylation, histone modifications, and ncRNA are crucial elements that regulate both oncogenes and tumor suppressor genes. In this review, we bring out the importance of the relationship between these epigenetics mechanisms and oxidative stress in cancer disease. Indeed, when an increase of oxidative stress occurs, different pathways are activated—among them, the regulation of epigenetic mechanisms, in order to activate antioxidant response. This happens to maintain a high metabolism, which is a characteristic of cancer cells, and also to avoid apoptosis induced by oxidative stress and genomic damage, for example, in the development of therapy resistance.

Specifically, the way in which miRNAs and lncRNAs regulate key oxidative stress genes is a promising horizon that we can approach to lead anticancer therapies in the future. An example of this is the miR-7/MAFG axis regulation in NSCLC and ovarian cancer. miR-7 methylation is able to modulate the antioxidant response over ROS production by *MAFG* downregulation. Furthermore, MAFG seems to be leading a methylator program of specific genes in colorectal and melanoma cancer, so that its increase could affect the methylation profile of different tumor suppressor genes, promoting the cancer cell survival. Interestingly, ROS can both trigger *MAFG* transcription, to respond to oxidative stress, but it also can promote the MAPK pathway hyperactivation that leads to the stabilization of MAFG, thus suggesting a new molecular mechanism that changes the epigenetic patterns through ROS/MAFG.

Advances and the development of new technologies would allow in the future targeting these elements that are critical for the cell. For example, the identification of aptamers that block MAFG function has opened new research directions for re-sensitizing cisplatin-resistant cells in vitro. Although other aptamers have been used in preclinical models to target cellular membrane receptors, targeting MAFG would be challenging, as it mainly located in the nucleus. Therefore, the development of new delivery approaches will be needed to use MAFG aptamers as a functional therapy. For example, the use of nanoparticles would be a powerful approach to deliver these inhibiting molecules, as it has been shown for microRNAs.

Although the current review focused on ncRNAs and their effectors in response to oxidative stress in solid tumors, it is important to highlight the crucial role of the other well-known epigenetic modifications of DNA and histones, as well as alterations of the enzymes that affect these modifications, not only in solid tumors but also in hematological malignancies. There are a substantial number of epigenetic modifiers that are approved for their use in cancer treatment [214,215]. Inhibitors of DNMTs, HDACs, or other chromatin modifier complexes, such as EZH2 inhibitors, have increased the variety of therapeutic approaches for the treatment of cancer. In addition, in the past years, the use of miRNA mimics in clinical trials [216,217], such as miR-29 or mir-34, for the treatment of cancer and other diseases has opened new possibilities for therapeutic strategies. In line with this, another potential approach to re-sensitize cisplatin-resistant cells would be to overexpress miR-7 agonists through nanoparticles. Although these approaches have yet to be studied, their feasibility has proven to be effective with other microRNAs.

Conclusively, understanding the relationship between oxidative stress and epigenetics mechanisms is crucial to both understanding cancer development and gaining insight into the cancer disease, to finally allow new innovative approaches through the use of specific therapeutic strategies.

## Figures and Tables

**Figure 1 antioxidants-09-00468-f001:**
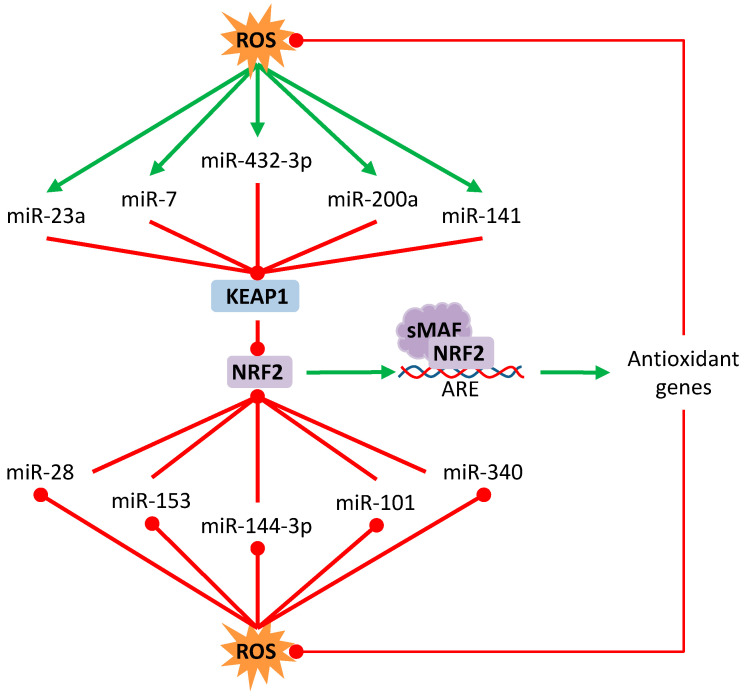
Regulation of NRF2/Keap1 by microRNAs. Oxidative stress can induce or repress some microRNAs that regulate the posttranscriptional expression of Keap1 or NRF2. Red lines with rounded end indicate inhibition. Green arrows indicate activation.

**Figure 2 antioxidants-09-00468-f002:**
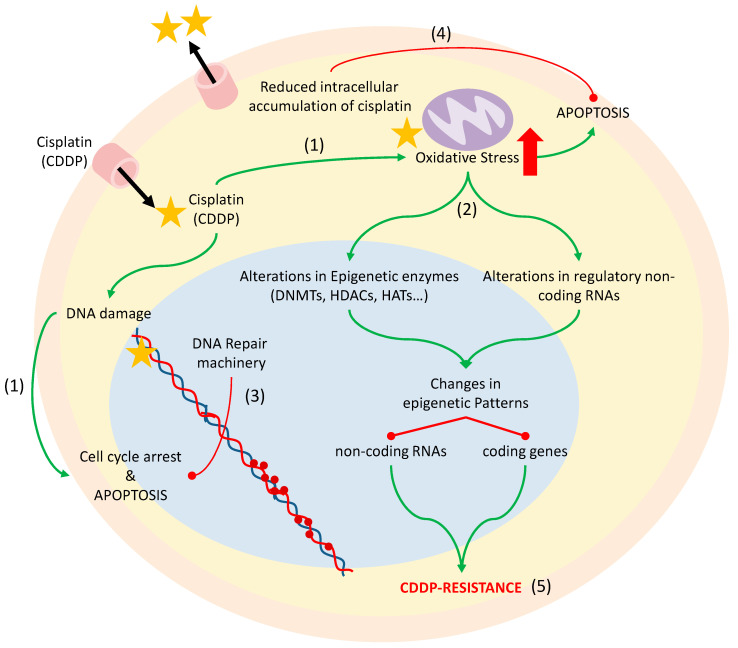
Effect of cisplatin inside the cell. Cisplatin is a platinum-derived drug that acts by generating aqueous species that bind to the N7 position in the guanine and causes inter and intra-strand cross-links in DNA that activate the mechanisms of cell cycle arrest or programmed cell death (1). In addition, cisplatin increases the oxidative stress of the cells, which leads to apoptosis (1), but also to alterations in the epigenetic enzymes and regulatory non-coding RNAs that change the epigenetic patterns (2). An increase of the DNA repair machinery (3), the reduced intracellular accumulation of cisplatin (4), or changes in the epigenetic machinery as a result of oxidative stress (5) lead to the development of cisplatin resistance and thus to the therapeutic limitation of its use. Red lines with rounded ends indicate inhibition. Green arrows indicate activation.

**Figure 3 antioxidants-09-00468-f003:**
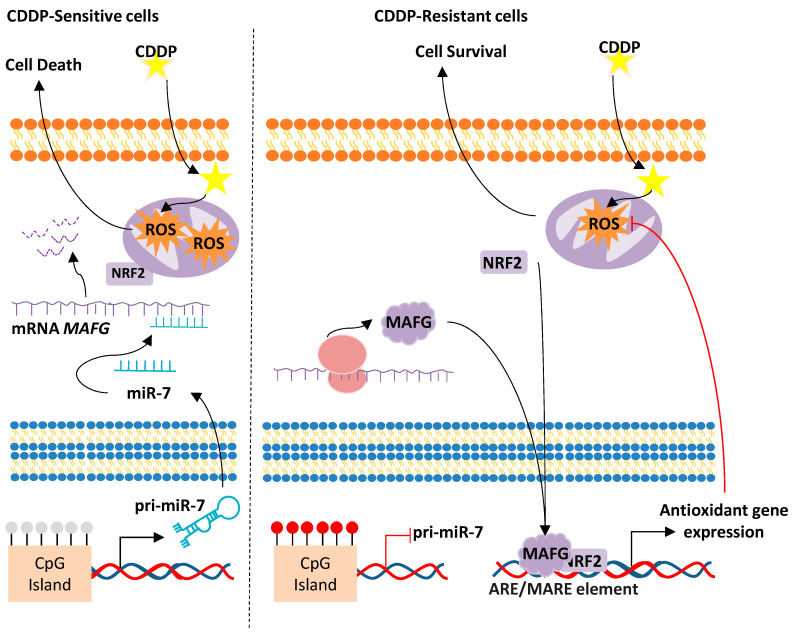
Proposed mechanism for the acquired resistance to cisplatin in NSCLC and ovarian cancer cells through miR-7 methylation and Musculoaponeurotic Fibrosarcoma Oncogene Family, protein G (MAFG) overexpression. Sensitive cells are represented on the left. The unmethylation of the miR-7 CpG Island allows its transcription, thus targeting the mRNA of MAFG, which promotes an increase of ROS and cell death. Cisplatin-resistant cells, right, harbor a methylated CpG island of miR-7, therefore allowing MAFG translation into protein, interaction with NRF2, and the transcription of antioxidant genes. CDDP, cisplatin; ROS, Reactive Oxygen Species.

**Table 1 antioxidants-09-00468-t001:** Relationship between the epigenetic modifications and the state of the chromatin.

Spot of Epigenetic Modification	Potentially Active Chromatin	Potentially Inactive Chromatin
DNA	DNA methylation outside CpG islands	DNA methylation in CpG islands of regulatory regions
Histones	Acetylated Unmethylated H3K4 Methylation	Deacetylated Methylated H3K4 unmethylation
Chromatin conformation	Open	Condensed Constitutive Heterochromatin

**Table 2 antioxidants-09-00468-t002:** MicroRNAs involved in the antioxidant response in cancer disease.

Name	Cancer TYPE	Target	Effect over Antioxidant Response	Reference
MIR-101	Breast	*NRF2*	downregulated	[141]
MIR-28	Breast	*NRF2*	downregulated	[147]
MIR-153	Breast	*NRF2*	downregulated	[148]
MIR-432-3P	Esophageal Squamous	*KEAP1*	upregulated	[142]
MIR-200A	Breast, Hepatocelullar, Esophageal squamous	*KEAP1*	upregulated	[144,145,146]
MIR-23A	Leukemic	*KEAP1*	upregulated	[149]
MIR-7	Neuroblastoma	*KEAP1*	upregulated	[143]
MIR-7	Non small cell lung	*MAFG*	downregulated	[150]
MIR-148B	Endometrial	*ERMP1*	downregulated	[151]
MIR-500A-5P	Breast	*TXNRD1* and *NFE2L2*	downregulated	[152]
MIR-143	Colon	*SOD1*	downregulated	[153]
MIR-139-5P	Breast	*MAT2A*	downregulated	[154]
MIR-29B	Ovary	*SIRT1*	upregulated	[155]
MIR-31	Oral squamous	*SIRT3*	downregulated	[156]
MIR-33A	Glioma	*SIRT6*	downregulated	[157]
MIR-517A	Melanoma	JNK sig. path.	downregulated	[158]
MIR-144-3P	Lung	*NRF2*	downregulated	[159]
MIR-340	Hepatocelullar	*NRF2*	downregulated	[160]
MIR-141	Ovary	*KEAP1*	upregulated	[161]

**Table 3 antioxidants-09-00468-t003:** Long non-coding RNAs (lncRNAs) regulated by the antioxidant response in cancer disease.

Name	Cancer TYPE	Intermediates/Effectors	Effect over Antioxidant Response	References
*NLUCAT1*	Lung cancer	HIF2A, NF-KB, NRF2	Upregulated	[162]
GABPB1-AS1	Hepatocellular carcinoma	GABPB1	Upregulated	[163]
SCAMP1	Renal cell carcinoma	miR-429/ZEB, JUN	Downregulated	[164]
HULC	Cholangiocarcinoma	Let-7a, Let-7b/IL-6	Upregulated	[165]
*H19*	Cholangiocarcinoma	miR-372, miR-373/CXCR4	Upregulated	[165]
*UCA1*	Non-small cell lung cancer	miR-495/NRF2	Upregulated	[169]
*UCA1*	Hepatocellular carcinoma	miR-184/OSGIN1	Upregulated	[170]

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
