# Peer review of "When Oxidative Stress Meets Epigenetics: Implications in Cancer Development"

_antioxidants, 2020, doi:10.3390/antiox9060468_

Round 1

Reviewer 1 Report

This review manuscript is to reviewed the correlation between oxidative stress and epigenetics in cancer development. The author reviewed the current researches of the effect of oxidative stress in the epigenetic machinery. The results are well organized. However, some points need to be addressed.

Major Point:

  1. Although the title of this review paper is “When Oxidative Stress Meets Epigenetics: Implications in Cancer Development”. But it seems the authors only focus DNA methylation in histone modification. In fact, there are other histone modification, such as methylation, phosphorylation, acetylation, ubiquitylation, and sumoylation. Some papers also mentioned that ubiquitylation and sumoylation involved during breast cancer progression with ROS. Thus, please add more information about other histone modification in line 77 “histone modifications” part.
  2. In line 90, the classification of ncRNAs should be more detail. The ncRNAs is wide known to classify to miRNA, siRNA, piRNA, and lncRNA. Please give a brief description about all these ncRNA and their correction with histone modification.
  3. The resolution of Figure 2 is poor, please replace it.

Reviewer 2 Report

In this review, Garcia et al. discuss about the effects of oxidative stress on the epigenetic machinery driving cancer onset and progression.

Overall, the topic is innovative and the review well organized.

I recommend to implement the review by citing the following work:

1 The role of epigenetic modifications in hematologic malignancies, including multiple myeloma,  shoud be also discussed (Expert Opinion Ther targets 2017 Jan;21(1):91-101. doi: 10.1080/14728222.2016.1266339);

2 Regarding miR-29, it is recommended the authors mention general reviews on its tumorigenic role: Oncotarget 2015 May 30;6(15):12837-61. doi: 10.18632/oncotarget.3805)

3. Regarding long non coding RNAs, the authors should report on the manuscript by Amodio et al., Leukemia 2018, which indicates that MALAT1 regulates NRF1 and NRF2 via Keap1 repression; moreover, they should discuss on the role of MALAT1 and EZH2 to repress miR 29b (Stamato et al.,  Oncotarget. 2017; 8:106527 106537. https://doi.org/10.18632/oncotarget.22507 )

Reviewer 3 Report

Garcí et al. have constructed a review article on the effect of oxidative stress in the epigenetic machinery and cancer progression. The article flows smoothly throughout. However, additional information seems required to make it more comprehensive. Some suggestions on the manuscript structure along with several minor edits that should be looked at are listed below.

Major concerns:

  1. Section 4.3. MiR-7/MAFG/ROS Axis as A Therapy-resistance mechanism. The author highlighted and emphasized the miR-7/MAFG/ROS Axis in a specific section. As one of the numerous miRNA-mediated therapy-resistance mechanisms described in this manuscripts, it seems more suitable to describe this portion in parallel with the other miRNAs in section 4.2. The section headings may need to rearrange accordingly. Also, in Figure 2, along with the miR-7/MAFG/ROS axis, addition of illustration of those miRNA-mediated treatment resistance mechanisms as described in line 330-348 will make it more comprehensive.
  2. Section 2. Implication of Epigenetics in the Development of Cancer. Only ncRNAs are touched. More specific information about DNA and chromatin modifications should be provided. Or refer the readers to some nice reviews on these topics.
  3. Line 152-163. As for “Alterations in lncRNAs contribute to the development and progression of human diseases”, only MALAT1 is mentioned. Please refer the readers to other comprehensive reviews on this field.
  4. Section 3. In addition to “Nrf2/Keap1”, other signaling pathways, such as MAPKs, NF-κB, PKC, STAT3, and PPARγ, which engaged in regulating pro-oxidant genes and antioxidant genes expression, and mediate cells oxidative injury and antioxidant defense system, need to be addressed.
  5. Section 4. The heading of “Resistance to Chemotherapeutic Treatment” seems to be segregated from other sections at the first sight. Please use alternative wordings to reflect the linkage of this portion with the aforementioned “oxidative stress” and “epigenetic changes” sections. Also, the discussion of “Chemotherapeutic Treatment” and “treatment resistance” seems to be centered on cisplatin only. Thus, the usage of “cisplatin-induced oxidative stress”, instead of“Treatment”, may be more specific and precise.

Specific comments:

  1. Figure 1. The descriptions of “DNA Methylation Coding Genes” and“DNA methylation Non-coding RNAs” is incomplete and confusing. Also, the locations of these two descriptions should be in the nucleus, along with “Changes in epigenetic patterns”. In addition, please check the usage of capital letters.
  2. Figure 2. The description of NRF2 in the figure should be added.
  3. Typos: Line 312: In fat – In fact; Line 332: past t years.

Author Response

Please, see attachment.

Round 2

Reviewer 1 Report

The authors have responsed all my comments and no further questions.

Reviewer 3 Report

After reorganization, the manuscript has been significantly improved. Overall, from my point of view, it is well-structured, easy to follow and now warrants publication.